# Mental Sampling in Multimodal Representations

**Jian-Qiao Zhu**
Department of Psychology
University of Warwick
j.zhu@warwick.ac.uk

**Adam N. Sanborn**
Department of Psychology
University of Warwick
a.n.sanborn@warwick.ac.uk

**Nick Chater**
Behavioural Science Group
Warwick Business School
nick.chater@wbs.ac.uk

## Abstract

Both resources in the natural environment and concepts in a semantic space are distributed "patchily", with large gaps in between the patches. To describe people's internal and external foraging behavior, various random walk models have been proposed. In particular, internal foraging has been modeled as sampling: in order to gather relevant information for making a decision, people draw samples from a mental representation using random-walk algorithms such as Markov chain Monte Carlo (MCMC). However, two common empirical observations argue against people using simple sampling algorithms such as MCMC for internal foraging. First, the distance between samples is often best described by a Lévy flight distribution: the probability of the distance between two successive locations follows a power-law on the distances. Second, humans and other animals produce long-range, slowly decaying autocorrelations characterized as $1/f$-like fluctuations, instead of the $1/f^2$ fluctuations produced by random walks. We propose that mental sampling is not done by simple MCMC, but is instead adapted to multimodal representations and is implemented by Metropolis-coupled Markov chain Monte Carlo ($MC^3$), one of the first algorithms developed for sampling from multimodal distributions. $MC^3$ involves running multiple Markov chains in parallel but with target distributions of different temperatures, and it swaps the states of the chains whenever a better location is found. Heated chains more readily traverse valleys in the probability landscape to propose moves to far-away peaks, while the colder chains make the local steps that explore the current peak or patch. We show that $MC^3$ generates distances between successive samples that follow a Lévy flight distribution and produce $1/f$-like autocorrelations, providing a single mechanistic account of these two puzzling empirical phenomena of internal foraging.

## 1 Introduction

In many complex domains, such as vision, motor control, language, categorization or common-sense reasoning, human behavior is consistent with the predictions of Bayesian models (e.g., [4, 39, 8, 3, 19, 25, 52, 54]). Bayes' theorem prescribes a simple normative method for combining prior beliefs with new information to make inferences about the world. However, the sheer number of hypotheses that must be considered in complex domains makes exact Bayesian inference intractable. Instead it must be that individuals are performing some kind of approximate inference, such as sampling [49, 38].

Using sampling to approximate Bayesian models of complex problems makes many difficult computations easy: instead of integrating over vast hypothesis spaces, samples of hypotheses can be drawn from the posterior distribution. The computational cost of sample-based approximations only scales with the number of samples rather than the size of the hypothesis space, though using a small number of samples results in biased inferences.

Interestingly, the biases in inference that are introduced by using a small number of samples match some of the biases observed in human behavior. For example, probability matching [49], anchoring effects [26], and many reasoning fallacies [10, 38] can all be explained in this way. However, there is as of yet no consensus on the exact nature of the algorithm used to sample from human mental representations.

Previous work has posited that people either use direct sampling or Markov chain Monte Carlo (MCMC) to sample from their posterior distribution over hypotheses [49, 26, 10, 38]. In this paper, we demonstrate that these algorithms cannot explain two key empirical effects that have been found in a wide variety of tasks. In particular, these algorithms do not produce distances between samples that follow a Lévy flight distribution, and separately they do not produce autocorrelations in the samples that follow $1/f$ scaling. A further issue is that mental representations have been shown to be "patchy" or multimodal – there are high probability regions separated by large regions of low probability – and MCMC is ill suited for multimodal distributions. We therefore evaluate one of the first algorithms developed for sampling from multimodal probability distributions, Metropolis-coupled MCMC ($MC^3$), and demonstrate that it produces both key empirical phenomena. Previously, Lévy flight distributions and $1/f$ scaling have been explained separately as the result of efficient search and the signal of self-organizing behavior respectively [48, 46], and we provide the first account that can explain both phenomena as the result of the same purposeful mental activity.

## 1.1   Distances between mental samples: Lévy flights

In the real world, resources are rarely distributed uniformly in the environment. Food, water, and other critical natural resources often occur in spatially isolated patches with large gaps in between. As a result, humans and other animals' foraging behaviors should be adapted to such patchy environments. In fact, foraging behavior has been observed to produce Lévy flights, which is a class of random walk whose step lengths follow a heavy-tailed power-law distribution [42]. In the Lévy flight distribution, the probability of executing a jump of length $l$ is given by:

$$P(l) \sim l^{-\mu} \tag{1}$$

where $1 < \mu \leq 3$, and the values $\mu \leq 1$ do not correspond to normalizable probability distributions. Examples of mobility patterns following the Lévy flight distribution have been recorded in Albatrosses [47], marine predators [43], monkeys [35], and humans [18].

Lévy flights are advantageous in patchy environments where resources are sparsely and randomly distributed because the probability of returning to a previously visited target site is smaller than in a standard random walk. In the same patchy environment, Lévy flights can visit more new target sites than a random walk does [6]. More formally, it has been proven that in foraging the optimal exponent is $\mu = 2$ regardless of the dimensionality of the space if (a) the target sites are sparse, (b) they can be visited any number of times, and (c) the forager can only detect and remember nearby target sites [48].

It has long been known that mental representations of concepts are also patchy [7] and remarkably the distance between mental samples also follows a Lévy flight distribution. For example, in semantic fluency tasks (e.g., asking participants to name as many distinct animals as they can), the retrieved animals tend to form clusters (e.g., pets, water animals, African animals) [45]. This same task has also been found to produce Lévy flight distributions of inter-response intervals (IRI) [37].

As we are interested in sampling, which can retrieve the same item multiple times, rather than destructive foraging, we conducted a new memory retrieval experiment. Ten native English speakers were asked to type animal names in English as they came to mind, explicitly allowing participants to revisit animal names. A detailed description of the experimental procedure can be found in the Supplementary Material and a summary of the data appears in Figure 2A: participants showed power-law scaling of their inter-retrieval intervals (IRI), replicating the main finding of [37]. IRIs can

be considered a rough measure of distance between samples, assuming that generating a sample takes a fixed amount of time, that there are unreported samples that are generated between each reported sample, and that the sampler has travelled further the more unreported samples that are generated. As further support, we used a standard technique from computational linguistics to measure the distances between mental samples, again finding Lévy flight distributions for these distances (see in the Supplemental Material for details and Table 1 for exponents).

## 1.2 Autocorrelations of mental samples: $1/f$ noise

Separate from investigations into the distances between mental samples, a number of studies have reported that many cognitive activities contain long-range, slowly decaying autocorrelations in time. These autocorrelations tend to follow a $1/f$ scaling law [24]:

$$C(k) \sim k^{-\alpha} \tag{2}$$

where $C(k)$ is the autocorrelation function of temporal lag $k$. The same phenomenon is often expressed in the frequency domain:

$$S(f) \sim f^{-\alpha} \tag{3}$$

where $f$ is frequency, $S(f)$ is spectral power resulting from a Fourier analysis and $\alpha \in [0.5, 1.5]$ is considered $1/f$ scaling.

$1/f$ noise is also known as pink or flicker noise, which varies in predictability intermediately between white noise (no serial correlation, $S(f) \sim 1/f^0$) and brown noise (no correlation between increments, $S(f) \sim 1/f^2$). Note that Lévy flights (i.e., randomly selecting a flight direction and then execute a flight distance that has power-law scaling as in Equation 1) are random walks and so produce $1/f^2$ noise instead of $1/f$ noise (see Supplementary Material for details).

$1/f$-like autocorrelations in human cognition were first reported in time estimation and spatial interval estimation tasks in which participants were asked to repeatedly estimate a pre-determined time interval of 1 second or spatial interval of 1 inch [17]. Subsequent studies have shown $1/f$ scaling laws in the response times of mental rotation, lexical decision, serial visual search, and parallel visual search [16], as well as the time to switch between different percepts when looking at a bistable stimulus (i.e., a Necker cube [12]).

Table 1: Empirical evidence for Lévy flights and $1/f$ noise in human mental samples

| Effect | Papers | Experiments | Main findings |
|--------|--------|-------------|---------------|
| Lévy flight | [37] | Memory retrieval task | Power-law exponents IRI $\mu \in [1.37, 1.98]$ |
| | Current | Memory retrieval task | Power-law exponents IRI $\mu \in [0.77, 2.39]$ |
| | | | Power-law exponents distance $\mu \in [0.76, 1.28]$ |
| $1/f$ noise | [17] | Time interval estimation | Power spectra slopes $\alpha \in [0.90, 1.20]$ |
| | | Spatial interval estimation | Power spectra slope $\alpha = 1$ |
| | [16] | Mental rotation | RT power spectra slope $\alpha = 0.7$ |
| | | Lexical decision | RT power spectra slope $\alpha = 0.9$ |
| | | Serial search | RT power spectra slope $\alpha = 0.7$ |
| | | Parallel search | RT power spectra slope $\alpha = 0.7$ |

## 2 Mental sampling algorithms

Given that the distances between mental samples follows a Lévy flight distribution and that the samples have $1/f$ autocorrelations (see Table 1 for summary), we now investigate which sampling algorithms can capture both aspects of human cognition.

We consider three possible sampling algorithms that might be employed in human cognition: Direct Sampling (DS), Random walk Metropolis (RwM), and Metropolis-coupled MCMC (MC$^3$). We define DS as independently drawing samples in accord with the posterior probability distribution. DS is the most efficient algorithm for sampling of the three, but can only be applied to relatively simple

examples as it often requires calculating intractable normalizing constants that scale exponentially with the dimensionality of the hypothesis space [28, 9]. DS has been used to explain biases in human cognition such as probability matching [49].

MCMC algorithms bypass the problem of the normalizing constant by simulating a Markov chain that transitions between states according only to the ratio of the probability of hypotheses [28]. We define RwM as a classical Metropolis-Hastings MCMC algorithm, which can be thought of as a random walker exploring the probability landscape of hypotheses, preferentially climbing the peaks of the posterior probability distribution [29, 21]. However, with limited number of samples, RwM is very unlikely to reach modes in the probability distribution that are separated by large regions of low probability. This leads to biased approximations of the posterior distribution [44, 38]. Random walks have been used to model clustered responses in memory retrieval [1], and RwM in particular has been used to model multistable perception [13], the anchoring effect [26], and various reasoning biases [10, 38]. However, RwM will struggle with multimodal probability distributions.

Our third algorithm is $MC^3$, also known as parallel tempering or replica-exchange MCMC, was one of the first algorithms to successfully tackle the problem of multimodality [14]. $MC^3$ involves running $M$ Markov chains in parallel, each at a different temperature: $T_1, T_2, ..., T_M$. In general, $1 = T_1 < T_2 < ... < T_M$, and $T_1$ is the temperature of the interest where the target distribution is unchanged. The purpose of the heated chains is to traverse valleys in the probability landscape to propose moves to far-away peaks (by sampling from heated target distributions: $\pi^{1/T}$), while the colder chains make the local steps that explore the current probability peak or patch. $MC^3$ decides whether to swap the states between two randomly chosen chains in every iteration [14]. In particular, swapping of chain $i$ and $j$ is accepted or rejected according to a Metropolis rule; hence, the name Metropolis-coupled MCMC

$$A^{swap} = \min\{1, \frac{\pi(x_j)^{1/T_i}\pi(x_i)^{1/T_j}}{\pi(x_i)^{1/T_i}\pi(x_j)^{1/T_j}}\} \qquad (4)$$

Coupling induces dependence among the chains, so each chain is no longer Markovian. The stationary distribution of the entire set of chains is thus $\prod_{i=1}^{M} \pi^{1/T_i}$ but we only use samples from the cold chain ($T = 1$) to approximate the posterior distribution [14]. Pseudocode for $MC^3$ is presented below. Note that $MC^3$ reduces to RwM when the number of parallel chains $M = 1$.

---
**Algorithm** Metropolis-coupled Markov chain Monte Carlo
---
1:    Choose a starting point $x_1$.
2:    **for** $t = 2$ to $L$ **do**
3:      **for** $m = 1$ to $M$ **do**             ▷ update all $M$ chains
4:        Draw a candidate sample $x' \sim \mathcal{N}(x_{t-1}^m, \sigma)$      ▷ Gaussian proposal distribution
5:        Sample $u \sim U[0,1]$
6:        $A^m = \min\{1, [\frac{\pi(x')}{\pi(x_{t-1}^m)}]^{1/T_m}\}$
7:        **if** $u < A^m$ **then** $x_t^m = x'$ **else** $x_t^m = x_{t-1}^m$ **end if**    ▷ Metropolis acceptance rule
8:      **end for**
9:      **repeat** floor($M/2$) **times**             ▷ swapping chains
10:        Randomly select two chain $i, j$ without repetition
11:        Sample $u \sim U[0,1]$
12:        $A^{swap} = \min\{1, \frac{\pi(x_t^j)^{1/T_i}\pi(x_t^i)^{1/T_j}}{\pi(x_t^i)^{1/T_i}\pi(x_t^j)^{1/T_j}}\}$
13:        **if** $u < A^{swap}$ **then** swap($x_t^i, x_t^j$) **end if**
14:      **end repeat**
15:    **end for**
---

## 3 Results

In this section, we evaluate whether the two key empirical effects of Lévy flights and $1/f$ auto-correlations can be produced by the Direct Sampling (DS), Random walk Metropolis (RwM), and Metropolis-coupled MCMC ($MC^3$) algorithms.

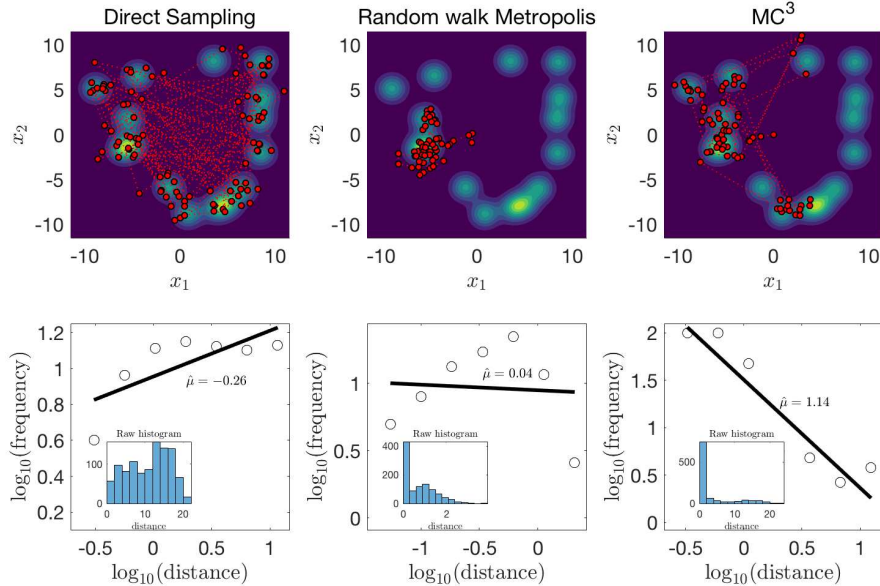

Figure 1: An example of searching behaviors in a 2D patchy environment. Each patch could represent a cluster of animal names. Repeated simulation of samplers in different environments can be found in Figure 2. **(Left Panel)** Simulation result for DS. The top panel shows the trajectory of first 100 positions (red dots). The bottom panel shows the log-log plot of flight distance distribution. The raw histogram of flight distance is also included in the bottom panel. The power-law exponent is fitted using LBN method, which corrects for irregular spacing of points [37]. **(Middle Panel)** the same treatments for RwM sampler. The Gaussian proposal distribution was an identity covariance matrix. **(Right Panel)** the same treatments for MC$^3$ sampler with 8 parallel chains and only the positions of the cold chain were displayed here. The Gaussian proposal distributions for all 8 chains had the same identity covariance matrix. For all three samplers considered here, only the first 1024 samples were used to match the length of human experiments.

## 3.1 Producing Lévy flights with sampling algorithms[1]

To simulate the sampling algorithms, we use a spatial representation of semantics (rather than the graph structure used in semantic networks), and we justify this choice in the Supplementary Material. For generality, we first focus on simulating patchy environments without making detailed assumptions about any one participant's semantic space. In particular, we create a series of 2D environment using $N_{mode} = 15$ Gaussian mixtures where the means are uniformly generated from $[-r, r]$ for both dimensions, where $r = 9$ and the covariance matrix is fixed as the identity matrix for all mixtures. This procedure will produce patchy environments (for example the top panel of Figure 1). We ran DS, RwM, and MC$^3$ on this multimodal probability landscape, and the first 100 positions for each algorithm can be found in the top panel of Figure 1. The empirical flight distances were obtained by calculating the Euclidean distance between two consecutive positions of the sampler. For MC$^3$, only the positions of the cold chain ($T = 1$) were used.

Power-law distributions should produce straight lines in a log-log plot. To estimate power-law exponents of flight distance, we used the normalized logarithmic binning (LBN) method as it has higher accuracy than other methods [37, 48]. In LBN, flight distances are grouped into logarithmically-increasing sized bins and the geometric midpoints are used for plotting the data. Figure 1 (bottom) shows that only MC$^3$ can reproduce the distributional property of flight distance as a Lévy flight with estimated power-law exponent $\hat{\mu} = 1.14$. Both DS ($\hat{\mu} = -0.26$) and RwM ($\hat{\mu} = 0.04$) produced values outside the range of power-law exponents found in human data. Indeed, RwM produces a highly non-linear log-log plot, differing in form as well as exponent from a Lévy flight. In the Supplemental Material, we support this result by showing how sampling from a low-dimensional

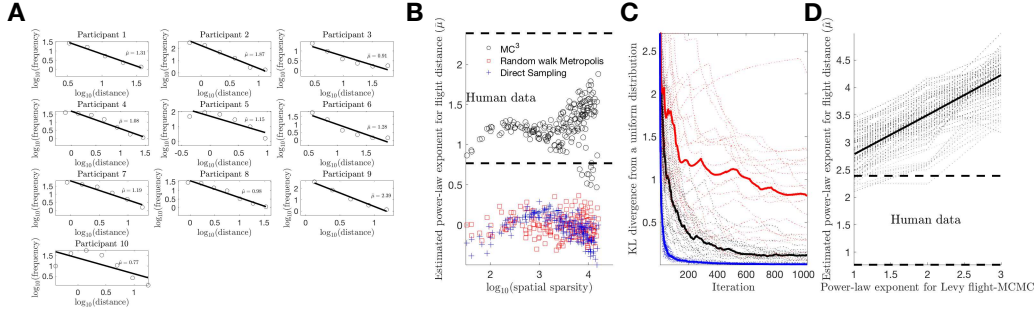

Figure 2: **(A)** Animal naming task as non-destructive mental foraging (10 participants). The estimated power-law exponents for IRI are $\mu \in [0.77, 2.39]$. **(B)** Estimated power-law exponents for flight distance distributions for the three sampling algorithms across different patchy environments, manipulating the spatial sparsity of the Gaussian mixture. The dashed lines show the range of power-law exponents suggested by our human data. Only MC$^3$ falls in this range. **(C)** KL divergence of mode visiting from the true distribution for the three sampling algorithms. Red denotes RwM, black denotes MC$^3$, and blue denotes DS. The patchy environments are the same for all three algorithms. The quicker the sampler approach zero KL divergence, the better the sampler is searching the patchy environment. The solid lines are medians of the dashed lines. **(D)** Simulated standard MCMC with power-law proposal distribution. The solid line shows the median in estimated power-law exponent. The dashed lines show the range of human data.

semantic space representation of animal names with MC$^3$ can produce Lévy flight exponents similar to those of produced by participants for distances.

Note that only one run of all three samplers in a patchy environment is shown in Figure 1. We also demonstrated the same samplers in different patchy environments where the impact of spatial sparsity on the estimated power-law exponents was investigated (see Figure 2B). In this simulation, the same number of Gaussian mixture were used but the range $r$ was varied: the higher $r$, the patchy environment is more likely to be sparse. The spatial sparsity was formally defined as the mean distance between Gaussian modes. With small or moderate spatial sparsity we found a positive relationship between spatial sparsity and the estimated power-law exponents for both DS and MC$^3$ (Figure 2B). In this range, only MC$^3$ produced power-law exponents in the range reported in our mental foraging task unlike DS and RwM. For all three algorithms, once spatial sparsity was too great only a single mode was explored and no large jumps were made.

We then vary values of hyperparameters and test whether this result is robust. In particular, we sampled 4 different values respectively for temperature spacing $\{0.5, 3, 7, 10\}$, number of parallel chains $\{2, 4, 6, 10\}$, resulting in 16 combinations of hyperparameters. Intuitively, larger temperature spacing, more parallel chains, and greater step size should lead to more explorative behavior of the sampler, and vice versa. Hence, for a certain environmental structure, MC$^3$ could tune these hyperparameters to balance between explorative and exploitative searches. For searches in semantic space of animal names, we run MC$^3$ repeatedly 10 times and the mean of these power-law exponents was considered. 62.50% hyperparameters reproduced Lévy flights.

We also checked whether MC$^3$ really is more suitable to explore patchy mental representations than RwM. In our simulated patchy environments, which used Gaussian mixtures with identity covariance matrix, an optimal sampling algorithm should visit each mode equally often, hence will produce a uniform distribution of visit frequencies over all the modes. To this end, the effectiveness of exploring such mental representation can be examined by computing a Kullback-Leibler divergence (KL) [28] between a uniform distribution over all modes and a the relative frequency of how often an algorithm visited each mode:

$$D_{KL}(\mathcal{H}_{1:t}||U) = \sum_{i=1}^{N_{mode}} \mathcal{H}_{1:t} \log \frac{\mathcal{H}_{1:t}}{1/N_{mode}} \qquad (5)$$

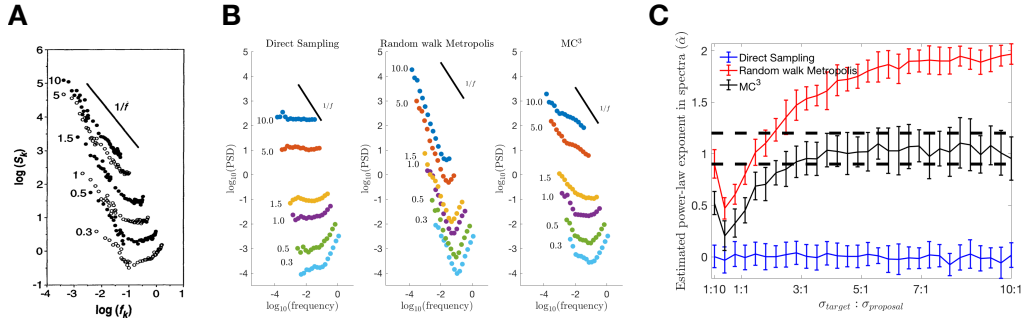

Figure 3: **(A)** Estimates of time duration show $1/f$ noise. The target durations for participants to estimate are shown next to scatterplots and the target duration ranged from 10s (top) to 0.3s (bottom). Best fit power-law exponents to the power spectra are $\alpha \in [0.90, 1.20]$ and this is also the range shown in dashed lines in Figure 3C. Figure adapted from [17] . **(B)** Power spectra produced by DS (left), RwM (middle), and MC³ (right). Only MC³ with 8 parallel chains can generate $1/f$ noise. For all the sampling algorithms, the first 1024 samples were used. **(C)** Estimated power-law exponent in power spectra are related to the ratio between Gaussian width and proposal step size. The power-law exponents for power spectra ($\hat{\alpha}$) were fitted following methods suggested by [17, 16]. The dashed lines show the range of $1/f^\alpha$ suggested by [17]. Error bars indicate $\pm$SEM. When the ratio is low, the acceptance rate of proposed sample should be low; it is the opposite case for the high ratio. The asymptotic behaviors of MC³ are $1/f$ noise, of RwM are brown noise, and of DS are white noise.

where $U$ is a discrete uniform distribution, $N_{mode}$ is the number of identical Gaussian mixtures, and $\mathcal{H}$ is the empirical frequency of visited modes up to time $t$. Samples were assigned to the closest mode when determining these empirical frequencies. The faster the KL divergence for an algorithm reaches zero, the more effective the algorithm is at exploring the underlying environment and the DS algorithm serves as a benchmark for the other two algorithms. As shown in Figure 2C, MC³ catches up to DS, while RwM lags far behind in exploring this patchy environment.

We checked whether the negative results for RwM were due to the choice of proposal distribution, by changing the Gaussian proposal distribution to a Lévy flight proposal distribution which has a higher probability of larger steps. Using a Lévy flight proposal distribution will straightforwardly produce power-law flight distance if the posterior distribution is uniform over the entire space (i.e., every proposal will be accepted). However, in a patchy environment, a Lévy flight proposal distribution will not typically produce a Lévy flight distribution of distances between samples that has estimated power-law exponents in the range of human data, as also can be seen in Figure 2D using different spatial sparsities. The reason for this is that the long jumps in the proposal distribution are unlikely to be successful: these long jumps often propose new states that lie in regions of nearly zero posterior probability.

## 3.2  Producing $1/f$ noise with sampling algorithms

A typical interval estimation task requires participants to repeatedly produce an estimate of the same target interval [17, 16]. For instance, participants were first given an example of a target interval (e.g., 1 second time interval or 1 inch spatial interval) and then repeatedly attempted to reproduce this target without feedback for up to 1000 trials. The time series produced by participants showed $1/f$ noise, with an exponent close to 1. However, the log-log plot of the human data is typically observed flatten out for the highest frequencies [17]. This effect has been explained as the result of two processes: fractional Brownian motion combined with white noise due to motor errors at the highest frequencies [17].

We investigated how well our three sampling algorithms can explain the autocorrelations in this temporal estimation task (Figure 3A: [17]). Gaussian distributions were used as target distributions for all sampling algorithms because the distribution of responses produced by participants was indistinguishable from a Gaussian [17]. For temporal estimation, it is known that the Gaussian distributions of responses have a scalar property that resembles Weber's law: the ratio of the mean to

the standard deviation is constant [34, 15]. For these simulations, we set this ratio between the mean and the standard deviation equal to 8 [34].

We then ran the sampling algorithms on the target durations tested by [17] (Figure 3B). Unlike in the simulations of distances between samples above, the time estimates produced by participants are estimates so we can directly compare them to the samples produced by the algorithms. RwM and $MC^3$ were initiated at the mode of the Gaussian distribution, and there was no burn-in period in our simulations. As in [17], for all three algorithms we added Gaussian motor noise to each sample to fit the upswing in the plot at higher frequencies. As each trial in the experiment started immediately as the previous trial ended, this resulted in the recorded estimate being equal to the sample plus the motor noise, but minus the motor noise from the previous trial, producing high frequency autocorrelations. Our motor noise had a constant standard deviation of $0.1$. Overall, the results show that only $MC^3$ produces $1/f$ noise ($\hat{\alpha} \in [0.5, 1.5]$), whereas DS tends to produce white noise ($\hat{\alpha} \in [0, 0.5]$) and RwM is closest to Brown noise ($1/f^2$:$\hat{\alpha} \in [1.5, 2]$).

RwM tends to generate brown noise because, if every proposed sample is accepted, then the algorithm reduces to first-order autoregressive process (i.e., AR(1)) [53]. This can be seen numerically by running the sampling algorithms using different ratios of the target distribution and proposal distribution standard deviations (Figure 3C). To see this relationship more clearly, in Figure 3C we did not add any motor noise. When the Gaussian width ($\sigma_{target}$) becomes much greater width of the Gaussian proposal distribution ($\sigma_{proposal}$), RwM produces brown noise. In contrast, $MC^3$ has a tendency to produce $1/f$ noise when the acceptance rate is high (Figure 3C black line). It has been shown that the sum of as few as three AR(1) processes with widely distributed autoregressive coefficients produces an approximation to $1/f$ noise [51]. As the higher-temperature chains can be thought of as very roughly similar to AR(1) processes with lower autoregressive coefficients, this may explain why the asymptotic behavior of the $MC^3$ is $1/f$ noise.

Note that, from effective sample size perspective, DS is clearly the best among three sampling algorithms. The cognitive emission of $1/f$ noise is very suboptimal from a statistical standpoint as it produces a smaller effective sample size than the independent samples drawn by DS or the mild autocorrelations found in RwM. However, our sampling account provides a reason for why the mind would produce $1/f$ noise: these long-range autocorrelations need to be tolerated in order to retain the possibility of generating samples from far-reaching modes.

We did a similar robustness check for hyperparameter settings using the same 16 combinations as above. For search in representation of temporal interval, only the 10s target interval was considered as it shows least influence of motor noise in the power spectra (see Figure 3A). 43.75% parameters reproduced $1/f$ noise. Combined, 18.75% parameters reproduced both Lévy flights in the animal naming task and $1/f$ noise in the time estimation task.

## 4 Discussions

Lévy flights are advantageous in a patchy world, and have been observed in many foraging task with humans and other animals. A random walk with Gaussian steps does not produce the occasional long-distance jump as a Lévy flight does. However, the swapping scheme between parallel chains of $MC^3$ enables it to produce Lévy-like scaling in the flight distance distribution. Additionally $MC^3$ produces the long-range slowly-decaying autocorrelations of $1/f$ scaling. This long-range dependence rules out any sampling algorithm that draws independent samples from the posterior distribution, such as DS, since the sample sequence would have no serial correlation (i.e., white noise). It also rules out RwM because the current sample solely depends on the previous sample. Both of these results suggest that the algorithms people use to sample mental representations are more complex than DS or RwM, and, like $MC^3$, are instead adapted to sampling from multimodal distributions.

However, if people are adapted to multimodal distributions, their behavior appears not to change even when they are actually sampling from a unimodal distribution. In Gilden's experiments, the overall distribution of estimated intervals (i.e., ignoring serial order) was not multimodal, instead it was indistinguishable from a Gaussian distribution [17]. Assuming that the posterior distribution in the hypothesis space is also unimodal then it is somewhat inefficient to use $MC^3$ rather than simple MCMC. Potentially the brain is hardwired to use particular algorithms, or it is slow to adapt to unimodal representations because it is very difficult to know that a distribution is unimodal rather

than just a single mode in a patchy space. Of course, it could be that even if MC$^3$ is always used, that the number of chains or temperature parameters are adapted to the task at hand. Additionally, it may be that a cognitive load manipulation would reduce the number of available chains and thus reduce exploration, which is an interesting prediction to test in future work.

Previous explanations of scale-free phenomenon in human cognition such as self-organized criticality argue that $1/f$ noise is generated from the interactions of many simple processes that produce such hallmarks of complexity [46]. Other explanations assume that it is due to a mixture of scaled processes like noise in attention or noise in our ability to perform cognitive tasks [50]. These approaches argue that $1/f$ noise is a general property of cognition, and do not tie it to other empirical effects. Our explanation of this scale-free process is more mechanistic, assuming that they reflect the cognitive need to gather vital informational resources from multimodal probability distributions. While autocorrelations make samplers less effective when sampling from simple distributions, they may need to be tolerated in multimodal distributions in order to sample other isolated modes.

An avenue for future work is to consider how MC$^3$ might be implemented in the brain. Researchers have proposed a variety of mechanisms for how sampling algorithms could be implemented in the brain, and these mechanisms can account for many neural response properties [2, 5, 20, 23, 32, 41]. We are not aware of any implementations of MC$^3$ in particular, but other work has proposed how multiple chains could be implemented in neural hardware [41]. Adapting this existing multiple-chain scheme to implement MC$^3$ would require: 1) running the different chains at different temperatures, 2) tracking the cold chain for the output samples, and 3) implementing a mechanism for switching states (or equivalently switching temperatures) between chains.

While we have evaluated MC$^3$ for internal sampling, it is interesting to consider whether it might describe some aspects of external search as well. Eye movements have been shown to produce both Lévy flights and $1/f$ noise, and the areas of interest in natural images are certainly multimodal [36].

Of course, we do not claim that MC$^3$ is the only sampling algorithm that is able to produce both $1/f$ noise and Lévy flights. It is possible that other algorithms that deal better with multimodality than MCMC, such as running a single chain at different temperatures [31, 40] or Hamiltonian Monte Carlo [2, 11], could produce similar results. Future work will further explore which algorithms can match these key human data.

## Acknowledgements

JQZ was supported by China Scholarship Council. ANS was supported by The Alan Turing Institute under the EPSRC grant EP/N510129/1. NC was supported by ERC grant 295917-RATIONALITY, the ESRC Network for Integrated Behavioural Science [grant numbers ES/K002201/1 and ES/P008976/1], the Leverhulme Trust [grant number RP2012-V-022], and RCUK Grant EP/K039830/1.

## Footnotes

[1]relevant code can be found at Open Science Framework: https://osf.io/26xb5/

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

# A  Lévy flights do not generate $1/f$ noise

In a Lévy flight, the direction of the flight is selected at random but the flight distance is distributed according to the power law [42, 47]. In an one-dimensional space, whether to move to the left or right is selected with equal probability, then the flight distance $l \sim U^{-1/(\mu-1)}$ where $U$ is the uniform distribution on [0,1]. This procedure guarantees the distribution of flight distances follows power-law with exponent $\mu$.

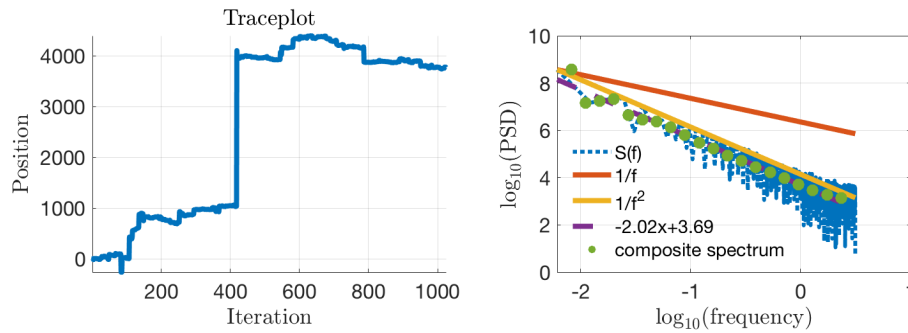

Figure 4: Autocorrelations produced by a Lévy flight. (**Left**) The traceplot of first 1024 locations of the Lévy flight. (**Right**) The power spectra of the locations.

In Figure 4, we simulated a Lévy flight and applied the same power spectra analysis on the traceplot that we did in the main text. Lévy flights produce independent increments so the location only depends on the previous location, and indeed the simulated Lévy flight produced $1/f^2$ noise ($\hat{\alpha} = 2.02$).

# B  Methods of the animal naming task

Ten native English speakers (6 Female and 4 Male, and aged 19-25 years) were recruited from the SONA system of Warwick University (Coventry, UK). The experiment lasted about 60 minutes or until the participants typed 1024 words. Participants sat in a soundproof cubicle for this task, and were paid £6 for the experiment.

