[Supplementary Material]


[46] G. C. Van Orden, J. G. Holden, and M. T. Turvey. Self-organization of cognitive performance. *Journal of Experimental Psychology: General*, 132(3):331, 2003.

[47] G. M. Viswanathan, V. Afanasyev, S. Buldyrev, E. Murphy, et al. Lévy flight search patterns of wandering albatrosses. *Nature*, 381(6581):413, 1996.

[48] G. M. Viswanathan, S. V. Buldyrev, S. Havlin, M. Da Luz, E. Raposo, and H. E. Stanley. Optimizing the success of random searches. *Nature*, 401(6756):911–914, 1999.

[49] E. Vul, N. Goodman, T. L. Griffiths, and J. B. Tenenbaum. One and done? Optimal decisions from very few samples. *Cognitive Science*, 38(4):599–637, 2014.

[50] E.-J. Wagenmakers, S. Farrell, and R. Ratcliff. Estimation and interpretation of $1/f^\alpha$ noise in human cognition. *Psychonomic Bulletin & Review*, 11(4):579–615, 2004.

[51] L. M. Ward. *Dynamical Cognitive Science*. MIT press, 2002.

[52] D. M. Wolpert. Probabilistic models in human sensorimotor control. *Human Movement Science*, 26(4):511–524, 2007.

[53] J. Xu and T. L. Griffiths. How memory biases affect information transmission: A rational analysis of serial reproduction. In *Advances in Neural Information Processing Systems*, pages 1809–1816, 2009.

[54] A. Yuille and D. Kersten. Vision as Bayesian inference: analysis by synthesis? *Trends in Cognitive Sciences*, 10(7):301–308, 2006.

## A  Lévy flights do not generate $1/f$ noise

In a Lévy flight, the direction of the flight is selected at random but the flight distance is distributed according to the power law [42, 47]. In an one-dimensional space, whether to move to the left or right is selected with equal probability, then the flight distance $l \sim U^{-1/(\mu-1)}$ where $U$ is the uniform distribution on [0,1]. This procedure guarantees the distribution of flight distances follows power-law with exponent $\mu$.

Figure 4: Autocorrelations produced by a Lévy flight. **(Left)** The traceplot of first 1024 locations of the Lévy flight. **(Right)** The power spectra of the locations.

In Figure 4, we simulated a Lévy flight and applied the same power spectra analysis on the traceplot that we did in the main text. Lévy flights produce independent increments so the location only depends on the previous location, and indeed the simulated Lévy flight produced $1/f^2$ noise ($\hat\alpha = 2.02$).

## B  Methods of the animal naming task

Ten native English speakers (6 Female and 4 Male, and aged 19-25 years) were recruited from the SONA system of Warwick University (Coventry, UK). The experiment lasted about 60 minutes or until the participants typed 1024 words. Participants sat in a soundproof cubicle for this task, and were paid £6 for the experiment.

The following instructions appeared on the screen before the task began:

> Hello Welcome!
>
> In this free association experiment, you are asked to type animal names as they come to mind. You will be shown the animal name you most recently reported on the screen and when you think of a different animal name, please type it into the computer.
>
> We are interested in the free association of animal names, so we would like you to report what new animal you are thinking of whenever the animal you are thinking of changes.
>
> It is okay to type in an animal name that you previously reported. Please let the experimenter know if you have any question before you begin.
>
> Press any key when you are ready to continue.

Participants were also told to press ENTER when finished typing an animal name. The inter-response interval (IRI) was the duration between last ENTER pressed and the very next key response.

## C Justifying a semantic space

Semantic representations are generally modelled with either a semantic space or a semantic network, and the algorithm that fits human data best can depend on the choice of representation [1, 22]. To test which representation is better to use for testing whether sampling algorithms can produce Lévy flights, we recruited two additional participants to complete a memory retrieval task similar to the animal naming task. However, in this task participants were allowed to report any noun as it came to mind, and not just animal names. Sampling algorithms using a semantic network should almost always predict IRI= 1 in this case, since almost all the nodes in the network can be a legal response. However a semantic space could still potentially produce power-law distributions of IRIs: under the simple assumption that the sampler reports the nearest noun, there can be many samples generated before the nearest noun changes.

Figure 5 shows that our two pilot participants produced power-law IRIs instead of constant IRIs. This relationship does not seem to hold for all of the IRIs, as the solid lines do not fit the data perfectly well, but we are most concerned with whether the longer IRIs follow a power-law distribution. When restricting our analysis to IRI $>$ 2s, the data do follow a power-law distribution as the dotted line fits the data well. This justifies our choice of a semantic space for this analysis.

Figure 5: Histogram of IRI (log-log plot) for two participants in noun recalling task. The estimated power-law exponents for tail distribution are $\hat{\mu} = 1.60$ (Participant 1) and $\hat{\mu} = 2.21$ (Participant 2).

## D Measuring the distance between samples in the animal naming task

To more directly investigate whether distances in a semantic space can be a good approximation of IRI, we mapped the animal names our participants produced into the 300-dimensional Word2Vec semantic space [30]. We first found the closest word (using the Ratcliff/Obershelp pattern-matching algorithm as implemented in the difflib.SequenceMatcher function in Python) within the Word2Vec dictionary for each participant response, as well as the animal terms identified by [45]. This resulted in 326 animal names with Word2Vec representations.

Figure 6: **(A)** 2D semantic space of all animal names. Each dot denotes one animal name. The contour represents a Gaussian mixture model on these animal names. **(B)** Histogram of flight distances for 10 participants from the animal naming task. The estimated power-law exponent $\hat{\mu} \in [0.76, 1.28]$. Median correlation coefficient between the flight distances and IRIs is 0.19. **(C)** Running three sampling algorithms on the Gaussian mixture model from **B**. As shown in the main text, only the $MC^3$ can replicate the power-law scaling of flight distance in the semantic space.

We assume that the representation of animals lies within some kind of manifold within the more general Word2Vec space, so in order to better represent the distances between animal names, we applied t-SNE to reduce the dimensionality of the space of the 326 animal names while respecting the manifold structure [27]. The perplexity parameter for t-SNE was set at 33 because this is the median category size of animal names suggested by [45]. The resultant 2D semantic space was shown in Figure 6A.

Using this 2D representation, we calculated the distances between successively reported animal names for all 10 participants, and found that the median correlation coefficient between flight distances and IRIs was 0.19, better than the median correlation coefficients we found for the 3D (0.04) or 4D (0.04) representations. Analyzing the distribution of distances between successive samples, we found they approximately have a power-law scaling (see Figure 6B). In this end, we chose to run our samplers in a 2D semantic space in the main text.

# E    Sampling in a semantic space

Using the low-dimensional representation we found in the previous section (see Figure 6A), we used a Dirchlet process Gaussian mixture model with the default parameters [33] to infer the probability distribution of animal names. The model found eight effective Gaussian components, and the mixture distribution is plotted as contours in Figure 6A. Our three candidate algorithms were run on the mixture distribution and the resulting power-law exponents were estimated (see Figures 6C). Only $MC^3$ produced exponents with the same sign as in the human data ($\hat{\mu} = 1.01$).