[Reviews · NeurIPS 2018]

Reviewer 1



One of the current speculative hypotheses in cognition is that the brain performs approximate Bayesian inference via some form of sampling algorithm. Based on this assumption, this paper explores which kind of Monte Carlo algorithm the brain might be using. In particular, previous work has proposed direct sampling (DS) from the posterior distribution, or random-walk MCMC. However, these two algorithms are unable to explain some empirically observed features of mental representations, such as the power law seen in the distance between consecutive, distinct “samples” (such as responses in a semantic fluency task) or, equivalently under certain assumptions, of the distribution of inter-response intervals. Also, autocorrelation of mental samples exhibit a specific “1/f” pattern. This paper argues that another type of MCMC algorithm, that is MC3 aka parallel tempering, which is a MCMC method designed to deal with multimodal, patchy posteriors (hence the “foraging” analogy), is instead able to explain these features. The authors demonstrate the better empirical validity of MC3 -- mostly as a proof of concept -- with a toy example and with real data obtained in a simple novel experiment. Comments: Quality: The paper is technically sound and, if we accept the whole “mental sampling” paradigm, presents an interesting hypothesis for how sampling is performed (i.e., via parallel processes -- chains -- that operate at different “temperatures”, and occasionally swap information). The current work is mostly a proof of concept in that the authors show general quantitative agreement with the data for some specific algorithm choices (e.g., temperature range, number of chains, proposal distributions). I think that a much broader exploration of the (hyper)parameter space is beyond the scope of this work (and perhaps pointless, since this is mostly a proof of existence), but it would be reassuring to see that these details do not substantially affect the results, at least for a couple of examples. Given the simplicity of the analysis, it should be straightforward to try out a few other algorithmic configurations. Importantly, the authors already show that RwM still does not reproduce features of the data when changing the proposal distribution to a heavy-tailed distribution, probably the most relevant control. However, a couple more examples would not hurt. Clarity: The paper is well-written and easy to follow throughout. Only found a couple of typos: line 111: follows → follow line 131: MCMC, was → MCMC, which was Originality: This work extends previous computational work in the cognitive sciences that proposed sampling (and specifically MCMC) as an algorithm used by the brain for approximating Bayesian inference. The novelty of this proposal is that it explains a couple of important features of observed data (Lévy flights and 1/f noise) via a MCMC method which was not previously considered as a model for cognition, as far as I know. Significance: There has been a trend in the past years of describing any chosen aspect of cognition via some specific modern (or less modern) machine learning technique, which is somewhat suspicious given that as humans we always had the tendency to describe the functioning of the brain -- once we started to understand its relevance for cognition -- with comparisons to the latest, fanciest engineering advancement. However, the current proposal is interesting in that some of the algorithmic features (e.g., the existence of different processes/chains, the associated temperatures) might be subject to direct experimental investigation, perhaps making this hypothesis testable beyond some qualitative agreement and proof of concept. In conclusion, this is a nice conference paper which presents an interesting idea and some preliminary support for it. After Author Feedback: Thank you for the clarifications. I find positive that the authors are going to share their code (I hope all the code to reproduce the experiments in the paper), and increases my belief in reproducibility of their results.

Reviewer 2



After reading the response: Thanks for the clarification on semantic networks; It is interesting that a walk on a semantic network would not predict the pattern because similar to a semantic space, a network can capture the similarity of nouns other than animals. It would be helpful if the authors clarify the assumptions underlying the representation of the semantic space and the required evaluation to back the main arguments of the paper: The assumption is that people's semantic space is patchy (which is backed up by previous work); the authors simulate a semantic space that has this property and show that their sampling method produces the desired levy-flight behavior. * The authors either need to show why the simulated representation is a good proxy of people mental space or use the human data (like the animal free recall) to create the semantic space. A comparison with different types of semantic representations (semantic space, topic model, etc) would be helpful. Does the choice of sampling algorithm interact with the semantic representation? * The authors need to evaluate the method on another dataset (to replicate human behavior); one example is the IRT pattern observed in the free recall data. For example, see Abbott el al (2015); Nematzadeh et al (2016). Summary and Quality Some of the patterns observed in human and animal (search) behavior suggest that the mental (or geographical) space have a patchy structure (which sometimes is a small world). More specifically, the authors argue that the distance between mental samples follows a heavy-tailed power-law distribution (levy flight); moreover, the autocorrelation between these samples follow a 1/f scaling law. They then study a few sampling algorithms (random walk, Metropolis-Hasting, and Metropolis-coupled MCMC). Their results suggest that only the last algorithm produces samples that satisfy both levy flight and autocorrelation properties. In the results section, the authors start with a simulated patchy environment but compare the results with human data. I do not understand the logic behind this analysis; more specifically, in Figure 2B, the authors compare the power-law exponents of the sampling algorithms on the simulated environment with human data. The authors acknowledge that the choice of the proposal distribution for Metropolis-Hasting affects the results and the two distribution they have examined does not predict the desired patterns. (Although they again compare the simulated environment with human data.) This observation -- if generalizable to a large number of proposal distributions) is interesting because it shows that having multiple chains (as in Metropolis-coupled MCMC) is important when sampling from patchy environments. The paper needs to better justify this claim. The paper seems to assume a vector-space representation for semantics (not a semantic network); the authors should explain the consequences of this decision; especially given that the previous work by Abbott et al shows that a random walk on a semantic network is enough to predict human search behavior. Clarity and significance The paper is mostly clear but the results section needs more details on what the examined space is and what human behavior is predicted. The appendix is not really used as a supplementary material; it is needed for understanding the paper. Parts of the appendix should be added to the paper such that it is self-contained. It is not clear to me what range of human behavior the suggested sampling algorithm can predict because the paper mostly focuses on a simulated environment. I am not sure how much this work tells us about mental sampling.

Reviewer 3



The authors ask how approximate inference could be implemented through some form of sampling scheme in humans. For complex domains it is required to entertain a large number of hypothesis in high dimensional spaces, in which regions of high probability may be separated by large regions of comparatively low probability. The main point of the manuscript is that previously proposed sampling algorithms do not result in 1/f autocorrelations between samples and do not result in distances between samples following a Lévy flight distribution. Here the authors propose to use the previously published MC^3 sampling algorithm, which couples several chains through the Metropolis acceptance rule but only uses the samples from one chain to estimate posteriors. Through simulations in part modeled after previous empirical experiments on human reasoning the authors show that MC^3 shows both the above properties, established empirically. Overall, the topic is relevant for the computational cognition/neuroscience audience at nips. I am quite surprised that the work by Orbán G, Berkes P, Fiser J, Lengyel M. or Laurence Aitchison, Máté Lengyel is neither mentioned nor discussed. There are no theorems here and no new deep network architectures, but I find the study relevant and interesting. Looking at figure 1 for the RwM algorithm, the authors should comment on how they chose parameters of the proposal distribution, e.g. the step size, in the chain because this looks as if a particularly disadvantageous choice was selected. The discussion lines 197-206 is helpful but limited. Similarly, how were the temperatures for MC^3 selected or was a parameter search carried out? How many free parameters are there in the different sampling schemes? [After Author Feedback] Thank you for the clarifications. I still think that this is a relevant and interesting contribution to NIPS.